

# Associations of moderate-to-vigorous physical activity with psychological problems and suicidality in Chinese high school students: a cross-sectional study

Mingli Liu[1,2,3], Jie Zhang[2,4], Kimberly E. Kamper-DeMarco[5], Elwin Hu[6] and Shuqiao Yao[3]

[1] Department of Psychology, Hunan University of Science and Technology, Xiangtan, Hunan, Country
[2] Department of Sociology, State University of New York Buffalo State, Buffalo, NY, USA
[3] Institute of Psychological Medicine, Second Xiangya Hospital, Central South University, Changsha, Hunan, China
[4] School of Sociology and Psychology, Central University of Finance and Economics, Beijing, Beijing, China
[5] Department of Psychology, State University of New York Buffalo State, Buffalo, NY, United States of America
[6] School of Psychology, Counselling and Psychotherapy, Cairnmillar Institute, Hawthorn East, Victoria, Australia

Corresponding authors
Mingli Liu, liumingli@hnust.edu.cn, mlliupsy@163.com
Shuqiao Yao, shuqiaoyao@csu.edu.cn

## ABSTRACT

**Background**. The body of literature regarding the associations between physical activity and psychological problems lacks consensus. Moreover, the role of gender has been scarcely investigated. The present study sought to fill the gap in the data by examining the associations, if any, between moderate-to-vigorous physical activity (MVPA), psychological problems, and self-harm behaviors based on different biological genders (male–female).

**Methods**. A self-administered questionnaire was used to measure MVPA and multiple psychological problems, including depression and anxiety; general emotion, behavior, and social problems; Attention Deficit Hyperactivity Disorder (ADHD); Oppositional Defiant Disorder (ODD); conduct problems; and self-harm behaviors. Participants were a nationally representative sample of Chinese high school students ($N = 13,349$). A multiple logistic regression analysis of MVPA, stratified by gender, was conducted on the risk of psychological problems and self-harm behaviors in the total sample.

**Results**. For boys, a high frequency of MVPA was associated with a lower risk of depression (OR = 0.68, 95% CI [0.57–0.81]) and anxiety (OR = 0.71, 95% CI [0.53–0.96]) compared to the reference group. The moderate frequency of MVPA was associated with a lower risk of ADHD (OR = 0.73, 95% CI [0.56–0.94]) compared to the reference group. For girls, both MVPA groups were associated with a lower risk of depression (moderate MVPA: OR = 0.81, 95% CI [0.70–0.94]; high MVPA: OR = 0.67, 95% CI [0.54–0.83]) compared to the reference group. High MVPA was associated with a lower risk of ADHD (OR = 0.58, 95% CI [0.37–0.89]) compared to the reference group. Only moderate frequency of MVPA was associated with a lower risk of ODD (OR = 0.79, 95% CI [0.64–0.97]).

**Conclusion**. MVPA was associated with a lower risk of depression, anxiety, ADHD, and ODD in a gender-specific and MVPA frequency-specific manner. This implies that for specific psychological issues, PA interventions that are modified based on gender

and frequency of MVPA may be more effective than PA interventions without these considerations.

## INTRODUCTION

It is well known in the literature that the occurrence of psychological problems rise sharply during adolescence (*Gore et al., 2011*; *Patton et al., 2012*). In fact, psychological problems are one of the leading causes of years of life lost in adolescents according to data from the World Health Organization's (WHO) Global Burden of Disease study (*Gore et al., 2011*; *Mokdad et al., 2016*). Furthermore, psychological problems during these years may influence health late in life (*Sawyer et al., 2012*). Therefore, psychological problems in adolescents are becoming a central public health concern (*Carli et al., 2014*; *Merikangas et al., 2010*). In order to create more appropriate prevention and intervention strategies, and subsequently decrease the burden of mental disorders, it is necessary to examine health risk factors for psychological problems in adolescents.

The literature has demonstrated that physical activity (PA) is not only strongly associated with physiological health benefits (*Buffart et al., 2014*), but is also associated with psychological health benefits, such as a reduction in depression and anxiety symptoms, increased self-esteem, social isolation, and even a decrease in suicidal ideation (*Vancampfort et al., 2017*; *Werneck et al., 2019*). For adolescents, it has been well documented that PA is significantly associated with a lower risk of depression (*Annesi, 2005*; *Daley et al., 2006*). Furthermore, PA intervention trials have significantly reduced depression, anxiety, psychological distress, and emotional disturbance in adolescence (*Ahn & Fedewa, 2011*; *Brown et al., 2013*). Although there is well established literature demonstrating the effects of PA on depression and anxiety, there are only a handful of studies reporting the associations of PA and other psychological problems, such as Attention Deficit Hyperactivity Disorder (ADHD; *Gapin et al., 2011*). However, the results from this body of literature remain inconsistent. For example, in one study, parent-rated ADHD was associated with low PA in adolescents with type 1 diabetes (*Nylander et al., 2018*). Recently, a study showed that increased moderate-to-vigorous PA (MVPA) was correlated with higher conduct problems in boys and hyperactivity in both genders (*Ahn et al., 2018*). It should be noted, however, that most of the studies mentioned above were highly heterogeneous regarding the participants and measurement instruments used (*Larun et al., 2006*). Therefore, further investigation is warranted and should be carried out in different population groups to determine whether associations exist between PA and adolescents' psychological problems. Furthermore, although PA seems to reduce depression and anxiety symptoms, little attention has been given to examining whether gender (i.e., male–female) influences psychological health outcomes. In a recent study (*Wilkinson et al., 2017*), anxiety related to body image was shown to be lower for American-born and Mexican-born females who

engaged in more PA than for those who were less engaged in PA. However, this relationship was different for the males in the study. Specifically, the association was significant for Mexican-born males but not for American-born males. On the other hand, a negative correlation was found between the frequency of PA and anxiety for both boys and girls among European adolescents (*McMahon et al., 2017*). These results demonstrate that different cultures and/or countries may influence the relationship between PA and anxiety.

According to a detailed analysis of internationally comparable data, the lowest rates of PA are mostly observed among juveniles who live in low-income and middle-income countries or regions (*Patton et al., 2012*). However, research concerning the association between PA and adolescents' psychological problems in these areas, especially large-scale population-based investigations, is scarce compared to studies in developed countries or regions. In China, there have been several studies that have investigated the associations between PA and depression, anxiety, hostility/aggression, and suicide ideation in adolescents. A study conducted in Bengbu, a city in southeastern China, found that junior high school students who performed vigorous PA (i.e., more than three days per week) had fewer depressive symptoms but these results were not replicated with anxiety (*Cao et al., 2011*). However, another study based in the same city suggested that although low-to-moderate PA was associated with fewer depressive and psychotic symptoms, vigorous PA was associated with both life satisfaction and self-esteem, as well as higher risk of general psychological health issues such as hostility/aggression, and suicidal ideation (*Tao et al., 2007*). Additionally, a study conducted in Xuzhou, a city in eastern China, revealed that minimal PA was associated with depression and anxiety in junior high school students (*Wang et al., 2014*). Because samples employed in the studies mentioned above were limited to junior high school students based on their respective cities, located in eastern China, these results cannot be generalized to the broader Chinese population. Thus, studies with large nationally representative samples of Chinese participants are lacking. In an attempt to remedy this shortcoming in the literature, one study was conducted across eight cities in China and revealed significant associations between PA (with moderate and vigorous PA measuresd separately) and reductions in depressive symptoms with pubertal stage examined as a moderator (*Sun et al., 2014*); however, this study neglected the potential confounding effect of other psychological problems and did not use recommended criteria for MVPA set by the WHO (*World Health Organization, 2010*) for child and adolescent psychological health.

The present study addressed some of these research gaps by analyzing the associations between different volumes of MVPA and multiple psychological problems in both junior and senior high school students across multiple cities in mainland China. These psychological problems include: depression; anxiety; general emotion, behavior, and social (GEBS) problems; ADHD; ODD; conduct problems; and suicidal ideation/self-harm behavior (hereafter, self-harm behaviors). Furthermore, given that gender has been proposed to moderate the relationship between PA and some psychological problems (i.e. *McMahon et al., 2017*; *Sun et al., 2014*; *Wilkinson et al., 2017*), we performed all analyses stratified by gender. We hypothesized that higher MVPA would be more protective against psychological health in boys than in girls.

## MATERIALS & METHODS

### Participants

High school pupils from 7th to 12th grade ($N = 13{,}349$; mean age: $15.18 \pm 1.90$) from ten urban areas (Beijing, Shanghai, Hangzhou, Guangzhou, Suzhou, Chengdu, Changsha, Shenyang, Yinchuan, and Langfang) were recruited. These areas represent different areas across mainland China. We used a stratified cluster sampling of all the public schools in these cities. First, we categorized all schools in each city into three groups based on their level of academic performance (low, medium, and high). Second, to make a representative sample of high school students, two to three medium-level high schools in each city were randomly chosen (totaling 23 schools) to take part in the survey. Finally, one to three classes of students within these schools were selected from each grade (7th through 12th grade) as the final sample of this study. More detailed information on these study procedures can be found elsewhere (see *Liu et al., 2016*; *Xin et al., 2016*). With the supervision of a research assistant, students completed the paper-based questionnaires individually in class. All participants in this study and their parents provided written informed consent. The Hunan Subjects Review Committee at Second Xiangya Hospital of Central South University provided ethical approval (No: CSMC-2009S167).

### Measurements

In this study, a self-administered questionnaire was used for the participants. It consisted of three sections: sociodemographic information, including age, gender, grade, BMI, and subjective socioeconomic status (SES), the Youth Risk Behavior Survey and questionnaires assessing psychological problems and self-harm.

The Youth Risk Behavior Survey questionnaire (*Brener et al., 2002*) evaluated students' frequency of MVPA (days/week). An example question is "During the past seven days, on how many days were you physically active enough that your heart beat fast and you breathed hard for a total of $\geq 60$ minutes?" In order to adjust for the potential confounding role of screen-based sedentary behavior (*Sun et al., 2014*), we also examined participants' screen time (ST) on a typical school day (hours/day). ST includes watching television, playing video or computer games, or using the computer for noneducational purposes. Although the recommendation of MVPA is 7 days/week, the percentage of adolescents who meet this criterion is very low, which indicates that Chinese high school students generally lack MVPA. Based on the frequency of MVPA, the participants were categorized into three groups: no MVPA (0 days/week), moderate frequency of MVPA ($\geq 1$ to $\leq 3$ days/week), and high frequency of MVPA (>3 days/week, i.e., 4–7 days/week). In line with previous studies, ST was categorized into four groups: no ST (0 hours/day), low ST ($\leq 1$ hours/day), moderate ST (>1 to <2 hours/day), and high ST ($\geq 2$ hours/day) (*Liu, Wu & Yao, 2016*). SES was reported. Scores ranged from one to ten. Low scores (i.e., 1,2,3) represented low SES while high SES, was represented by high scores (i.e., 7,8,9).

A battery of questionnaires evaluated participants' psychological symptoms and self-harm behaviors. Depressive symptoms were examined using the Centers for Epidemiologic Studies-Depression Scale (CES-D, *Radloff, 1991*; *Wang et al., 2013*). The CES-D has 20 items, and responses range from 0 (rarely) to 3 (most of the time). Anxiety symptoms

were measured using the Multidimensional Anxiety Scale for Children (MASC *March et al., 1997*; *Yao et al., 2007*). The MASC has 39 items, and responses range from 0 (never applies to me) to 3 (often applies to me). GEBS problems, ADHD symptoms, ODD, and conduct problems were assessed with the Youth Self-Report (YSR; *Achenbach, Dumenci & Rescorla, 2001*; *Yao et al., 2009*). The YSR consists of 112 items divided into nine subscales with responses ranging from 0 (not true) to 2 (very true). Finally, self-harm behaviors were examined using a 5-item subscale from the Health-Risk Behavior Inventory for Chinese Adolescents (HBICA; *Wang et al., 2012*). Responses range from 0 (never) to 4 (always). For all these scales, high scores indicate higher levels of the corresponding symptoms. Cronbach's alphas for all of the scales and subscales were good, ranging from 0.71 to 0.95.

Data was dichotomized based on the different cutoff scores (separated by gender) for each psychological problem(i.e., depression (CES-D score $\geq$ 22 for boys and $\geq$ 24 for girls (*Primack et al., 2009*; *Roberts, Lewinsohn & Seeley, 1991*); anxiety (T-score of MASC $\geq$65; March, 2007); GEBS problems (above the 90th percentile; (*Achenbach, Dumenci & Rescorla, 2001*)); ADHD symptoms (YSR-DSM-Oriented subscale, T-score $\geq$ 65; (*Achenbach, Dumenci & Rescorla, 2001*; *Achenbach & Rescorla, 2001*) ODD (YSR-DSM-Oriented subscale, T-score $\geq$ 65); conduct problems (YSR-DSM-Oriented subscale, T-score $\geq$ 65); and self-harm behaviors (HBICA subscale, above the 90th percentile; (*Wang, 2012*)). Specifically, individuals were identified as either meeting criteria for a psychological problem (1) or not (0).

## Statistical analysis

To compare the characteristics of each MVPA frequency group to sociodemographic variables, psychological problems and self-harm behaviors, F-tests were used for the continuous variables, and chi-square tests were used for the categorical variables. The effect size was estimated by $\eta^2$ for ANOVA or $\varphi/\upsilon$ for chi-square tests. All tests and analyses were examined by gender.

To evaluate the associations of MVPA frequency with psychological problems and self-harm behavior, multiple logistic regression analyses of the effect of MVPA frequency on the risk of psychological problems and self-harm behavior, adjusted for age, gender, grade (grade 7–9 or 10–12), BMI, SES, and ST, were conducted first for the total sample and then stratified by gender using MVPA frequency as the independent variable and each psychological problem or self-harm behavior as the dependent variable. Odds ratio (ORs) and 95% confidence intervals (CIs) were used as the measurement values of the associations between MVPA frequency and psychological problems; with no MVPA being used as the reference group. For the regression models, all exposure variables were fully observed. However, there were different missing data percentages for the outcomes (depression, 2.9%; anxiety, 1.9%; GEBS problems, 7.2%; ADHD, 10.3%; ODD, 8.8%; conduct problems, 11.0%; and self-harm behavior, 0.7%). A value of $p < 0.05$ (two-tailed) was used for statistical significance across all analyses. All statistical analyses were performed using the software of SPSS 19.0 and STATA 12.0.

## RESULTS

### Characteristics of the participants and MVPA

The proportion of boys and girls in the sample was 50.7% and 49.3%, respectively. The mean age (SD) of all participants was 15.18 (1.90) years and more than fifty percent (50.1%) of all participants were in junior high school. Across all participants, the mean BMI (SD) was 19.61 (3.35), and the mean SES score (SD) was 6.13 (1.64). The Pearson $\chi^2$ results indicated that MVPA frequency was significantly associated with television time ($p < 0.001$ for both gender) and playing video/computer games or using the computer for noneducational purpose time ($p = 0.011$ for boys; $p < 0.001$ for girls).

High school students' characteristics involving the three levels of MVPA (i.e., no, moderate frequency, high frequency) significantly differed in all sociodemographic variables and ST. The participant characteristics of each MVPA frequency (stratified by gender) are reported in Table 1. For boys, more than thirty percent (33.1%) reported no MVPA in the last seven days, 42.0% reported moderate (i.e., one to three days) MVPA in a week, and 24.9% reported high (i.e., more than three days) MVPA in a week. Sociodemographic and ST (both time spent watching television and playing video/computer games) for boys were significantly different ($p < .05$) across all three levels of MVPA with effect sizes ranging from 0.01 to 0.32.

As for girls, approximately 49% reported having no MVPA in a week, 36.8% reported moderate (i.e., one to three days) MVPA in a week, and 14.2% reported high (more than three days) MVPA in a week. Sociodemographic variables and both ST types differed significantly ($p < 0.001$). The effect sizes for these differences ranged from 0.02 to 0.47.

### Distribution of psychological problems and self-harm behavior

The prevalence of psychological problems and self-harm behavior, stratified by gender and MVPA frequency, are reported in Table 2.

For boys, the risk for self-harm behavior and the examined psychological problems, with the exception of ODD ($p = 0.141$) and conduct problems ($p = 0.631$), differed across MVPA levels ($p < 0.05$). The effect sizes ranged from 0.04 to 0.10. For girls, the risk for the examined psychological problems, with the exception of conduct problems ($p = 0.122$), differed across MVPA levels ($p < 0.05$). The effect sizes for these differences ranged from 0.04 to 0.08. No significant difference in the risk prevalence of self-harm behavior was found across MVPA levels ($p = 0.674$).

### Associations between MVPA, psychological problems and self-harm behavior

The results of the adjusted ORs and the corresponding 95% CIs of the logistic regression analyses for each psychological problem and self-harm behavior regarding MVPA in the total sample are reported in Table 3.

Engaging in high MVPA was associated with a lower risk of depression (OR = 0.67, 95% CI [0.59–0.77]), anxiety (OR = 0.75, 95% CI [0.60–0.94]), and ADHD (OR = 0.76, 95% CI [0.60–0.98]) compared to the reference group (i.e., no MVPA). Moderate MVPA was associated with a lower risk of depression (OR = 0.86, 95% CI [0.78–0.95]), ADHD

Liu et al. (2020), *PeerJ*, DOI 10.7717/peerj.8875

**Table 1 Characteristics of participants by MVPA level.[a]**

| Variables | Total | boys (days/week) (N, %) | | | | girls (days/week) (N, %) | | | |
|---|---|---|---|---|---|---|---|---|---|
| | | 0 (2,236; 33.1%) | ≥1 to ≤3 (2,842; 42.0%) | >3 (1,687; 24.9%) | ES; p | 0 (3,224; 49.0%) | ≥1 to ≤3 (2,425; 36.8%) | >3 (935; 14.2%) | ES;p |
| Age, mean (SD) | 15.18 (1.90) | 15.78 (1.82) | 15.06 (1.86) | 14.58 (1.75) | 0.06;<0.001 | 15.91 (1.76) | 14.54 (1.83) | 14.20 (1.59) | 0.15;<0.001 |
| Junior grade[b], N (%) | 6,836 (50.1) | 697 (31.2) | 1,549 (54.5) | 1,231 (73.0) | 0.32;<0.001 | 824 (25.6) | 1,611 (66.4) | 789 (84.4) | 0.47;<0.001 |
| BMI, mean (SD) | 19.61 (3.35) | 20.24 (3.69) | 20.00 (3.63) | 19.95 (3.66) | 0.01;0.028 | 19.48 (2.83) | 18.87 (3.00) | 18.52 (2.77) | 0.02;<0.001 |
| SES[c], mean (SD) | 6.13 (1.64) | 5.85 (1.67) | 6.17 (1.63) | 6.44 (1.68) | 0.02;<0.001 | 5.92 (1.56) | 6.21 (1.58) | 6.56 (1.65) | 0.02;<0.001 |
| Exposure levels of television (hours/school day), N (%) | | | | | | | | | |
| 0 | 7,869 (58.6) | 1,349 (60.4) | 1,394 (49.2) | 836 (49.6) | 0.09;<0.001 | 2,182 (67.8) | 1,454 (60.0) | 594 (63.6) | 0.06;<0.001 |
| | 3,903 (29.1) | 617 (27.6) | 1,029 (36.3) | 553 (32.8) | | 742 (23.1) | 710 (29.3) | 230 (24.6) | |
| >1 to ≤2 | 935 (7.0) | 142 (6.4) | 265 (9.3) | 136 (8.1) | | 161 (5.0) | 166 (6.9) | 61 (6.5) | |
| >2 | 710 (5.3) | 124 (5.6) | 148 (5.2) | 161 (9.5) | | 133 (4.1) | 92 (3.8) | 49 (5.2) | |
| Exposure levels of video or computer games or using a computer for non-educational purpose (hours/school day), N (%) | | | | | | | | | |
| 0 | 9,290 (70.2) | 1,491 (67.6) | 1,834 (65.7) | 1,060 (64.8) | | 2,418 (75.7) | 1,705 (71.4) | 714 (77.9) | |
| ≤1 | 2,440 (18.4) | 407 (18.5) | 589 (21.1) | 324 (19.8) | 0.04;0.011 | 513 (16.1) | 473 (19.8) | 120 (13.1) | 0.05;<0.001 |
| >1 to ≤2 | 761 (5.8) | 142 (6.4) | 191 (6.8) | 104 (6.4) | | 135 (4.2) | 137 (5.7) | 47 (5.1) | |
| >2 | 736 (5.4) | 164 (7.4) | 178 (6.4) | 149 (9.1) | | 130 (4.1) | 74 (3.1) | 35 (3.8) | |

**Notes.**

Abbreviations: MVPA, moderate to vigorous physical activity; BMI, body mass index; SES, subjective economic status; ES, effect size.

[a]Percentages do not always equal 100 because of rounding. Values may not always sum to sample size because of missing data

[b]Junior grade indicates 7th–9th grade.

[c]SES was considered as continuous variables (ranging from 1-the worst, to 10-the best).

Liu et al. (2020), *PeerJ*, DOI 10.7717/peerj.8875

**Table 2  Prevalence (%)[a] of psychological problems by MVPA levels.**

| Variables | Total N (%) | Boys (days/week) N (%) | | | | Girls (days/week) N (%) | | | |
|---|---|---|---|---|---|---|---|---|---|
| | | 0 | ≥1 to ≤3 | >3 | ES; p | 0 | ≥1 to ≤3 | >3 | ES; p |
| Depression, N (%) | 3,539 (27.3) | 729 (34.3) | 779 (28.5) | 369 (22.7) | 0.10; <0.001 | 924 (29.1) | 558 (23.3) | 180 (19.6) | 0.08;<0.001 |
| Anxiety, N (%) | 943 (7.2) | 181 (8.8) | 199 (7.5) | 89 (5.7) | 0.05; 0.002 | 258 (8.3) | 164 (6.9) | 52 (5.7) | 0.04;0.019 |
| GEBS problems, N (%) | 1,264 (10.2) | 244 (12.3) | 241 (9.1) | 139(8.8) | 0.05; <0.001 | 351 (11.7) | 216 (9.5) | 73 (8.0) | 0.05;0.001 |
| ADHD, N (%) | 766 (6.4) | 151 (7.9) | 142 (5.6) | 93 (6.2) | 0.04; 0.006 | 225 (7.7) | 121 (5.5) | 34 (3.9) | 0.06;<0.001 |
| ODD, N (%) | 1,352 (11. 1) | 236 (12.1) | 273 (10.4) | 186 (12.0) | 0.03;0.141 | 355 (12.0) | 212 (9.4) | 90 (10.0) | 0.04;0.007 |
| Conduct problems, N (%) | 867 (7.3) | 137 (7.3) | 169 (6.7) | 97 (6.5) | 0.01;0.631 | 232 (8.0) | 179 (8.2) | 53 (6.1) | 0.03;0.122 |
| Self-harm behavior, N (%) | 1,431 (10.8) | 280 (12.7) | 285 (10.2) | 167 (10.1) | 0.04;0.007 | 340 (10.6) | 266 (11.1) | 93 (10.1) | 0.01;0.674 |

**Notes.**

Abbreviations: MVPA, moderate to vigorous physical activity; GEBS, general emotion, behavior, and sociality; ADHD, Attention deficit/hyperactivity disorder; ODD, oppositional defiant disorder; ES, effect size.

[a]Percentages do not always equal 100 because of rounding. Values may not always sum to sample size because of missing data.

[b]Total prevalence of oppositional defiant problems was 11.1% for more than 3% samples' *T* score was between 64.5 to 65 (cutoff: *t*-score ≥ **64.50**).

**Table 3  Logistic regression analysis of MVPA on the risk of psychological problems and self-harm behavior in the total sample.**

| Variables | Self-reported MVPA levels (days/week) | |
|---|---|---|
| | ≥1 to ≤3 | >3 |
| Depression, OR (95% CI) , p | 0.86 (0.78 to0.95), 0.003 | 0.67 (0.59 to 0.77), <0.001 |
| Anxiety, OR (95% CI), p | 0.94 (0.80 to 1.11), 0.447 | 0.75 (0.60 to 0.94), 0.014 |
| GEBS problems, OR (95% CI), p | 0.90 (0.78 to 1.04), 0.157 | 0.86 (0.70 to 1.04), 0.121 |
| ADHD, OR (95% CI), p | 0.76 (0.63 to 0.91), 0.004 | 0.76 (0.60 to 0.98), 0.031 |
| ODD, OR (95% CI), p | 0.86 (0.74 to 0.99), 0.041 | 0.99 (0.83 to 1.19), 0.922 |
| Conduct problems, OR (95% CI), p | 1.03 (0.86 to 1.23), 0.748 | 0.84 (0.66 to 1.06), 0.145 |
| Self-harm behavior, OR (95% CI), p | 0.96 (0.84 to 1.11), 0.613 | 0.85 (0.77 to 1.11), 0.417 |

**Notes.**

Abbreviations: MVPA, moderate to vigorous physical activity; GEBS, general emotion, behavior, and sociality; ADHD, Attention deficit/hyperactivity disorder; ODD, oppositional defiant disorder; OR, adjusted odds ratio; CI, confidence interval.

Adjusted variables: age, gender, junior grade, body mass index, subjective economic status, volume of television and video games/computer use for non-education purpose.

(OR = 0.76, 95% CI [0.63–0.91]), and ODD (OR = 0.86, 95% CI [0.74–0.99]) compared to the reference group.

The detailed results of the adjusted ORs and the corresponding 95% CIs of the logistic regression analyses for each psychological problem and self-harm behavior regarding MVPA volumes stratified by gender are reported in Table 4.

For boys, high MVPA was associated with a lower risk of depression (OR = 0.68, 95% CI [0.57–0.81]) and anxiety (OR = 0.71, 95% CI [0.53–0.96) compared to the reference group. Moderate MVPA was associated with a lower risk of ADHD symptom (OR = 0.73, 95% CI [0.56–0.94]) compared to the reference group. For girls, both high and moderate MVPA were associated with a lower risk of depression (moderate MVPA: OR = 0.81, 95% CI [0.70–0.94]; high MVPA: OR = 0.67, 95% CI [0.54–0.83]) compared to the reference group. High MVPA was associated with a lower risk of ADHD symptom (OR = 0.58, 95% CI [0.37–0.89]) compared to the reference group. Only a moderate MVPA was associated with a lower risk of ODD (OR = 0.79, 95% CI [0.64–0.97]). No significant association was found between MVPA and self-harm behavior or other psychological problems for either gender.

## DISCUSSION

The present study examined the potential benefits of MVPA on psychological health in Chinese adolescents. In this large population-based investigation, a higher frequency MVPA was associated with less depression regardless of gender. ADHD and ODD symptoms decreased based on gender and the frequency of MVPA. Importantly, the various associations of MVPA with anxiety, ADHD, and ODD in both boys and girls indicated that gender and frequency of MVPA play a role in the associations between MVPA and psychological problems. These results support recommendations from the WHO concerning MVPA in adolescents (*World Health Organization, 2010*).

Liu et al. (2020), *PeerJ*, DOI 10.7717/peerj.8875

Peerʃ

**Table 4  Logistic regression analysis of associations of psychological problems and self-harm behavior with MVPA by gender.**

| Variables | Self-reported MVPA levels (days/week) | | | |
|---|---|---|---|---|
| | boys ($N = 5651$) | | girls ($N = 5512$) | |
| | $\geq 1$ to $\leq 3$ | >3 | $\geq 1$ to $\leq 3$ | >3 |
| Depression, OR (95% CI) , p | 0.88 (0.77 to 1.01), 0.068 | 0.68 (0.57 to 0.81), <0.001 | 0.81 (0.70 to 0.94), 0.005 | 0.67 (0.54 to 0.83), <0.001 |
| Anxiety, OR (95% CI), p | 0.94 (0.75 to 1.18), 0.604 | 0.71 (0.53 to 0.96), 0.024 | 0.94 (0.74 to 1.19), 0.588 | 0.85 (0.59 to 1.21), 0.358 |
| GEBS problems, OR (95% CI), p | 0.84 (0.68 to 1.04), 0.1034 | 0.85 (0.66 to 1.10), 0.211 | 0.96 (0.78 to 1.19), 0.702 | 0.87 (0.64 to 1.20), 0.400 |
| ADHD, OR (95% CI), p | 0.73 (0.56 to 0.94), 0.016 | 0.88 (0.65 to 1.20), 0.427 | 0.81 (0.62 to 1.06), 0.125 | 0.58 (0.37 to 0.89), 0.014 |
| ODD, OR (95% CI), p | 0.92 (0.75 to 1.12), 0.392 | 1.12 (0.89 to 1.42), 0.330 | 0.79 (0.64 to .97), 0.025 | 0.84 (0.63 to 1.12), 0.240 |
| Conduct problems, OR (95% CI), p | 1.00 (0.77 to 1.30), 0.995 | 0.92 (0.67 to 1.26), 0.595 | 1.04 (0.82 to 1.33), 0.734 | 0.74 (0.51 to 1.06), 0.100 |
| Self harm, OR (95% CI), p | 0.87 (0.72 to 1.06), 0.176 | 0.92 (0.72 to 1.16), 0.456 | 1.01 (0.83 to 1.24), 0.892 | 0.92 (0.69 to 1.23), 0.589 |

**Notes.**

Abbreviations: MVPA, moderate to vigorous physical activity; GEBS, general emotion, behavior, and sociality; ADHD, Attention deficit/hyperactivity disorder; ODD, oppositional defiant disorder; OR, adjusted odds ratio; CI, confidence interval.

Adjusted variables: age, junior grade, body mass index, subjective economic status, volume of television and video games/computer use for non-education purpose, Gender-matched no MVPA groups served as reference groups.

Participants engaging in no MVPA during the previous week were observed in 33.1% of boys and 49% of girls. It is worth noting that the present study's rates of no MVPA for females is higher than ratings reported in a previous study conducted by Sun and colleagues (2014), which measured PA using moderate PA (at least 20 min each day) or vigorous PA (at least 30 min each day). Moreover, *Sun et al. (2014)* revealed that 42.8% of girls reported no vigorous PA, and 40.6% of girls reported no moderate PA. This disparate finding is reasonable given that we used the WHO daily recommended amount of MVPA (60 min per day) for children and adolescents (*World Health Organization, 2010*). Furthermore, we found that only 24.9% of boys and 14.2% of girls had a high frequency of MVPA, indicating that there is a low rate of high school students who have enough MVPA in mainland China. We also found a significant difference between the three different MVPA frequencies and age for these junior high school students. For both genders, younger students engaged in more MVPA. Furthermore, a significantly different prevalence of psychological problems based on the frequency of MVPA and gender was found. After adjustment, logistic regression analyses revealed different associations between individual psychological problems, self-harm behavior and MVPA for each gender. For male and female adolescents, the risk of depression decreased consistently with more MVPA. The effects of the frequency of MVPA are in line with the findings of previous studies (*Ahn & Fedewa, 2011*; *Annesi, 2005*; *Brown et al., 2013*; *Cao et al., 2011*; *Tao et al., 2007*).

Surprisingly, in the present study, a high frequency of MVPA was associated with less anxiety in boys, but not in girls. The significant association found in boys in this study is similar to the findings in another Chinese study and international research (*McDowell, MacDonncha & Herring, 2017*; *McMahon et al., 2017*; *Tajik et al., 2017*; *Wang et al., 2014*). However, a lack of consensus remains given that the present study's results contradict studies conducted in an eastern inner province of China (*Cao et al., 2011*; *Tao et al., 2007*), which suggested no significant associations between PA and anxiety in high school students. These differences could be explained by how previous studies defined PA, as well as participants' sociodemographic factors. The participants in the aforementioned studies were recruited solely from one province in China rather than across multiple provinces and centers, as was the case in the present study. Thus, the present findings are more generalizable to all Chinese adolescents. More importantly, these studies did not stratify the analysis of the associations by gender.

In the present study, we did not find significant associations between MVPA and anxiety in girls, and only high MVPA was associated with less anxiety in the total sample. This result is also different from the study by Wilkinson and colleagues (2017), which suggested that lower social physique anxiety was associated with more PA per day for American and Mexican girls and Mexican-born boys but not in American-born boys. However, for European adolescents, a negative correlation was found between the frequency of PA and anxiety for both boys and girls (*McMahon et al., 2017*). This discrepancy might be explained by a culture-specific effect. In China, given that traditional masculine gender-role stereotyping is pervasive across society, this may provide justification for boys to take part in PA. In this study, the higher prevalence of MVPA (at least one day/week) for boys (66.9%) compared to girls (51.0%) also supports this claim. This gender-role stereotyping may

provide boys with more psychological rewards, which in turn, may produce anti-anxiety effects (*Brunes, Gudmundsdottir & Augestad, 2015*; *Salmon, 2001*). On the other hand, in China, the traditional feminine gender-role stereotype dictates that girls should be gentle and non-muscular. As a result, performing any form of PA may go against cultural norms and evoke unpleasant feelings. In addition, in the study by *McMahon et al. (2017)*, although a negative association was found between the frequency of PA and anxiety for both genders, at the highest levels of PA, higher anxiety was associated with a greater frequency of PA in girls. This finding leaves open the question of whether PA and anxiety differs across gender, with the possibility of negative effects of PA on anxiety for girls. The existing evidence regarding gender differences in the association between PA and anxiety is limited and as such, more research is needed to clarify this issue.

This study is the first to examine the association between MVPA and ADHD, ODD, and conduct problems in Chinese adolescents. For MVPA and ADHD, significant associations were found in both genders. These results are consistent with the finding by *Nylander et al. (2018)*, who found that ADHD was associated with low PA in adolescents with type I diabetes. Interestingly, gender influences this association. For boys, moderate MVPA was associated with fewer ADHD symptoms compared with no MVPA. On the other hand, only high MVPA was associated with fewer ADHD symptoms in girls compared to no MVPA. Regarding ODD, only a moderate frequency of MVPA was associated with fewer ODD symptoms in girls compared to no MVPA. The mechanisms that influence these gender differences are complex and one possibility is that they are influenced by cultural and social norms (*Ahn et al., 2018*). Alternatively, these outcomes may reflect gender differences in coping mechanisms when dealing with psychological problems (*Nolen-Hoeksema, 1987*). For example, *Nolen-Hoeksema (1987)* noted that boys are more likely to engage in distracting behaviors such as PA, which in turn, inhibits their negative moods while girls are more likely to amplify their moods with ruminative responses about their state of discomfort. In turn, these approaches could lead to a higher sensitivity of the therapeutic effects of PA in girls. The present study's results support this hypothesis. Furthermore, given the masculine stereotyping and gender-roles within Chinese society, MVPA behaviors are conceivably more appealing to Chinese male adolescents. Therefore, it is probable that only a high frequency of MVPA may exert a therapeutic effect on negative moods for Chinese male adolescents.

The present study focused on the frequency of MVPA for adolescents recommended by the *World Health Organization (2010)* and comprehensively considered multiple psychological problems experienced by adolescents. The findings extend previous research in this field. The current study also adds important insight into the potential role of gender on the relationship between MVPA and depression, anxiety, ADHD, and ODD, which indicates that MVPA might be an effective role in reducing psychological problems. This may provide evidence for the development of policies or recommendations regarding appropriate levels of MVPA for juveniles in mainland China, and potentially other countries or regions. Additionally, unlike previous Chinese studies, which are not representative of the broader Chinese population, the recruitment method in the present study enables a better generalizability of the results to all Chinese adolescents. This is because the

participants were recruited from major cities in North, South, East, West and Central China. Notwithstanding the strengths, there are also some limitations to consider. First, this study employs a cross-sectional observational design. Therefore, we cannot address the issue of causality of the associations between MVPA and psychological problems and self-harm behavior. Second, although the data analyses were adjusted for potential confounding variables, it is not possible to exclude the potential biases induced by other factors. For instance, the pressure of academic performance may lead to less PA and greater anxiety at the same time. In addition, self-reported MVPA and psychological problems and self-harm behavior may cause measurement errors, which would subsequently influence the results. Finally, despite the better generalizability of the results than that of previous Chinese studies, the present study did not recruit participants from rural areas or other cities in China. Future Chinese studies should endeavor to explore the effects of PA on psychological problems and self-harm behavior among rural Chinese adolescents.

## CONCLUSIONS

The present study provides empirical support for the recommendations put forth by the WHO. Additionally, this study contributes to the body of literature by informing researchers, health practitioners and policymakers of the psychological health benefits brought upon by PA. Overall, for mainland Chinese high school students, the level of MVPA determined by the WHO was associated with a lower risk of depression, anxiety, ADHD, and ODD. These associations were influenced by gender and the frequency of MVPA. Ultimately, the results indicate that PA interventions targeting specific psychological issues require understanding of how gender and frequency of MVPA may influence the therapeutic effects amongst Chinese adolescents.

## ACKNOWLEDGEMENTS

The authors would like to acknowledge the help of the following individuals for contributing to participants recruitment and measurement at the study centers: Wenbing Gao (Chinese Academy of Sciences), Wei Hong and Jing Liu (Peking University), Zhengyan Jiang (Zhejiang University), Yanqing Tang (China Medical University), Jin Jing (Sun Yat-Sen University), Wenqing Fu (Suzhou University), Yi Huang (Sichuan University), and Jianqun Fang (Ningxia Medical University).

## ADDITIONAL INFORMATION AND DECLARATION

### Funding

This work was supported by the China National Key Technologies R&D Program in the 11th 5-year plan of China (No. 2009BAI77B02), the Hunan Provincial Natural Science Foundation of China (No. 2017JJ3082), and the Hunan Provincial Social Science Foundation of China (No. 16YBA154). The funders had no role in study design, data collection and analysis, decision to publish, or preparation of the manuscript.

## Grant Disclosures

The following grant information was disclosed by the authors:
China National Key Technologies R&D Program: 2009BAI77B02.
Hunan Provincial Natural Science Foundation of China: 2017JJ3082.
Hunan Provincial Social Science Foundation of China: 16YBA154.

## Competing Interests

The authors declare there are no competing interests.

## Author Contributions

- Mingli Liu conceived and designed the experiments, performed the experiments, analyzed the data, prepared figures and/or tables, authored or reviewed drafts of the paper, and approved the final draft.
- Jie Zhang and Shuqiao Yao conceived and designed the experiments, prepared figures and/or tables, authored or reviewed drafts of the paper, and approved the final draft.
- Kimberly E. Kamper-DeMarco performed the experiments, analyzed the data, authored or reviewed drafts of the paper, and approved the final draft.
- Elwin Hu performed the experiments, authored or reviewed drafts of the paper, and approved the final draft.

## Human Ethics

The following information was supplied relating to ethical approvals (i.e., approving body and any reference numbers):

The Hunan Subjects Review Committee at Second Xiangya Hospital of Central South University approval to carry out the study within its facilities (Ethical Application: CSMC-2009S167)

## Data Availability

The raw measurements are available in a Supplementary File.

## Supplemental Information

Supplemental information for this article can be found online at http://dx.doi.org/10.7717/peerj.8775#supplemental-information.

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
