# Peer review of "Associations of moderate-to-vigorous physical activity with psychological problems and suicidality in Chinese high school students: a cross-sectional study"

_PeerJ, doi:10.7717/peerj.8775_

## Round 0.1 · original submission · Major Revisions

Thank you for submitting this interesting paper. However, the paper has major issues that need to be addressed before I can further process it. Please revise it accordingly.

Reviewer 1 ·

Basic reporting

Even not being a native English speaker, I found several inconsistencies in the grammar. Therefore, I strongly recommend another grammar revision.

Experimental design

No comment

Validity of the findings

No comment

Additional comments

Major

Overall: even not being a native English speaker, I found several inconsistencies in the grammar. Therefore, I strongly recommend another grammar revision.

The authors should avoid the term “psychopathologies” , as they did not evaluated this in the present study.

Even with an extensive literature review in the introduction, the literature gaps that the present article will focus are not clear, please revise this.

The authors should better describe the sampling process, it is not clear in the current version.

It is not clear why the authors adopted the cutoff point of 3 days/week with 60 min of MVPA, it is lower than the recommendations (7d/wk) and even the widely used 300 min/wk.

The authors should avoid the use of dichotomic p-values, it is not necessary to insert p for trend of OR values (it is possible to infer potential differences analyzing 95%CI). It is also possible to infer potential associations using 95%CI from OR values.

The authors did not analyze the sex-differences that they presented as an important objective. For this, it would be necessary to insert interaction terms of sex in the analysis, the authors should revise this.

I am not sure if it is necessary to repeat the results of Table 3 using a figure.

Minor

The title “Gender, physical activity, psychopathologies, and suicidality in Chinese students” is not in line with the manuscript. I suggest something in the line of: Sex-differences in the association between physical activity…

- The authors should better describe the methods on abstract. How MVPA was collected? Which indicators of psychological problems did the authors adopted? Statistics?

- The authors should insert informative numbers in the results of abstract, p-values are not informative and its dichotomic interpretation should be avoided.

- The authors should re-write the conclusion of abstract (it is not clear).

Line 69: There is a more recent paper showing this (GBD 2018)

·

Basic reporting

This is an inetresting study that examined the association of physical activity and psychological problems in high school students in 10 Chinese cities. The strength include relatively large sample size, valid measurements, analyses stratified by gender, including psychological problems other than depression and anxiety (for instance, ADHD). However, there are several major limitations.
1. Language is a problem and proof reading is needed.
2. The representativeness of the sample is doubtful. Sampling method should be described in detail. All participants are from 10 cities, therefore the results are not generalizable in rural students or other cities in China.
3. There are important potential confounders that have not been addressed in this study. For example, the pressure on academic performances is a well-known stress in high school students. The pressure may lead to less physical activity and anxiety at the same time.
4. Line 214, 'The participants' gender was not significantly different'. I cannot understand this sentence.
5. As shown in table 2, it seems that the prevalence of depression in male were higher than those in female, across all three MVPA groups. Please explain this.
6. The authors used 'prevalence rate' in line 233 and 278. However, prevalence is not a rate. Please use 'prevalence'.
7. Using self-report data is a limitation, particularly when the authors did not report test-retest reliability. Please mention this in the limitation.
8. It would be helpful to examine the association between MVPA and psychological problems combining both genders.

Experimental design

no comment

Validity of the findings

no comment

Additional comments

no comment

---

## Round 0.2 · Minor Revisions

I am also satisfied with the revisions, however, the paper still has some language issues. I suggest the authors to edit the paper again.

Reviewer 1 ·

Basic reporting

None.

Experimental design

None.

Validity of the findings

None.

Additional comments

None.

·

Basic reporting

no comment

Experimental design

no comment

Validity of the findings

no comment

Additional comments

My concerns have been successfully addressed by the authors. I have no further comments.

---

## Round 0.3 · accepted · Accept

No further comments. Thank you for your revisions.